# Technological Optimization of the Stirrup Casting Process with the Use of Computer Simulations

**DOI:** 10.3390/ma15196781

**Published:** 2022-09-30

**Authors:** Marcin Małysza, Robert Żuczek, Dorota Wilk-Kołodziejczyk, Krzysztof Jaśkowiec, Mirosław Głowacki, Piotr Długosz, Piotr Dudek

**Affiliations:** 1Łukasiewicz Research Network–Krakow Institute of Technology, Zakopiańska 73, 30-418 Krakow, Poland; 2Faculty of Metals Engineering and Industrial Computer Science and Faculty of Computer Science, Electronics and Telecommunications, AGH University of Science and Technology, Al. Mickiewicza 30, 30-059 Krakow, Poland; 3Division of Computer Science, Faculty of Mathematics and Natural Science, Jan Kochanowski University of Humanities and Sciences in Kielce, Stefana Żeromskiego 5, 25-369 Kielce, Poland; 4NOWOZNAL Non-Ferrous Pressure Die-Casting Department, ALPHA TECHNOLOGY sp. z o.o. sp.k., ul. Eugeniusza Kwiatkowskiego, 132-005 Niepołomice, Poland

**Keywords:** high pressure die casting, computer simulation, ANSYS, Flow3D, stirrup

## Abstract

The article presents the optimization of high-pressure die casting process technology for equestrian stirrups with the application of computer simulation. In the initial stage, the output technology was analyzed, and on the basis of a series of virtual experiments the cause of defects in the casting was highlighted. The optimization process includes different designs of a gating system. Additionally, the casting application properties were evaluated in an exploitation simulation, taking into account predicted defects resulting from the casting and solidification process. Based on the conducted analyses, technological changes were made to the casting technology design allowing the defects occurring in the original technological concept to be removed.

## 1. Introduction

The use of computer simulation in the design and manufacturing of cast parts is nowadays a non-separable element. Both commercial and proprietary software solutions are widely used. An example of such an application is the work [1], which presents the use of a computer simulation of casting to reduce defects resulting from shrinkage of the liquid metal during solidification. The elimination of this type of defects has a very significant impact on the quality of the casting. During normal operation in foundry conditions that kind of defect can be identified only after the production is completed. The use of simulation methods helps in the verification and optimization of the structure and allows minimizing waste. The simulation result was compared with the experimental test, thus verifying the precision of the virtual experiments. In another article [2], the problem of residual stresses was identified and resolved at design stage. Such action may lead to the development and application of appropriate process control methods during production in order to reduce premature destruction of the casting in exploitation conditions. The work [3] presents the production process of the casting of A365 alloy by the EPSC-VL method, in which liquid metal fills a layered ceramic mold in the environment of reduced pressure. The presented comparative analysis of computer simulation and real industrial trials shows the link between research and the production process. Another issue presented in [4] is multi-scale modeling (MCA — Modified Cellular Automaton). The use of the cell automaton method allows predicting the evolution of the microstructure, taking into account heterogeneous nucleation and redistribution of elements in the liquid and solid phases and at the phase boundary. Based on the experimental data, a nucleation model was proposed to predict the grain structure, as well as the dendritic microstructure evolution during the solidification process. The grain morphology of the test casting and cast airplane turbine wheel in different settings was predicted and compared with the experimental results. Another publication [5] is concerned with the development of a heat transfer model during the cooling process of steel billets and coupling it with the microsegregation effect. Parameters such as the type of nozzle, arrangement, distance between the nozzles and the plate, as well as the spray range and stream intensity, were taken into account. Rather than the natural convection, thermal radiation and contact cooling of the individual rolls were calculated. Another issue using computer simulations was described in [6], where the methodology of predicting final mechanical properties in pressure die casting was described. Computer simulations of high-pressure die casting process include process parameters, i.e., velocity profile of the plunger, thermal profile of the die mould, ventilation and cooling. Research work describing mentioned conditions [7,8,9] shows how computer simulations can help in solving process issues. For example, very important parameters describing the liquid metal velocities can be optimized by pQ^2^ chart. Additionally, the influence of air evacuation is presented as a very important parameter of whole casting method. Optimization of the velocity factor additionally allows for lowering the costs. Better knowledge of process parameters helps to produce high quality castings.

The presented works describing various casting technologies prove the wide range of application and the connection of computer simulations, not only in research but also for industrial practice.

This publication refers to the manufacturing process of stirrups which are a part of the harness used in horse riding. The stirrups can have different shapes and can be manufactured of various materials. In Figure 1, a few examples of stirrups are presented, i.e., steel stirrups subjected to an oxidation process to improve the appearance, stirrups of forged steel and stirrups with a detaching element.

Despite hundreds of years of using stirrups and much redesigning, their structure still keeps its original assumptions. The stirrup is connected to the saddle through the stirrup leather. It is important in reaction to the force transmitted through the entire system, starting with the work of the system itself -> rider -> stirrup. The system of forces and reactions in the rider’s knee joint depends on the type of seat and the height of the stirrup [11]. The research shows what load the stirrup must carry along with its cooperating parts. This is described in the Maquet load model, which includes a description of the forces acting on the knee. Such an example has been described in [12] by presenting different values depending on the driving method. The value calculated by the authors can reach an average of approximately 120% of the rider’s body weight and, at the peak value, of 162% of the rider’s body weight. This short presentation of stirrups as an element of equipment aims to show how important this invention is, which has been with mankind for hundreds of years.

## 2. Materials and Methods

### 2.1. Simulation of Casting Process

The presented manuscript concerns the improvement of the quality of equestrian stirrup casting using the high-pressure die method. Virtual testing of the mould cavity filling process takes into account the movement of a stream of liquid metal, air evacuation from the mould cavity volume and solidification. The tests were carried out in the Flow-3D software. The software uses the proprietary TrueVOF method, which allows very high accuracy to analyze the behavior of the free surface in the simulation of liquid metal flow through the gating system. The tested liquid alloy flow is described by the Navier–Stokes equation, Equation (1).
(1)$$\begin{array}{l} \frac{\partial u}{\partial t}+\frac{1}{V_{F}}\left\{uA_{x}\frac{\partial u}{\partial x}+vA_{y}R\frac{\partial u}{\partial y}+wA_{z}\frac{\partial u}{\partial z}\right\}-\varepsilon \frac{A_{y}v^{2}}{xV_{F}}=-\frac{1}{\rho }\frac{\partial p}{\partial x}+G_{x}+f_{x}-b_{x}-\frac{R_{SOR}}{\rho V_{F}}\left(u-u_{w}-\partial u_{s}\right)\\ \frac{\partial v}{\partial t}+\frac{1}{V_{F}}\left\{uA_{x}\frac{\partial u}{\partial x}+vA_{y}R\frac{\partial u}{\partial y}+wA_{z}\frac{\partial u}{\partial z}\right\}+\varepsilon \frac{A_{y}uv}{xV_{F}}=-\frac{1}{\rho }\left(R\frac{\partial p}{\partial y}\right)+G_{y}+f_{y}-b_{y}-\frac{R_{SOR}}{\rho V_{F}}\left(u-u_{w}-\partial u_{s}\right)\\ \frac{\partial w}{\partial t}+\frac{1}{V_{F}}\left\{uA_{x}\frac{\partial u}{\partial x}+vA_{y}R\frac{\partial u}{\partial y}+wA_{z}\frac{\partial u}{\partial z}\right\}=-\frac{1}{\rho }\frac{\partial p}{\partial z}+G_{z}+f_{z}-b_{z}-\frac{R_{SOR}}{\rho V_{F}}\left(u-u_{w}-\partial u_{s}\right) \end{array}$$
*U* = (*u*, *v*, *w*)—fluid velocity; *P*—pressure; *G*—gravitational acceleration; *t*—viscosity tensor of effort; *K U*—braking coefficient;*RSOR U/r*—acceleration caused by the injected mass of aluminum at zero speed; *F*—other force. 

After filling the die cavity, the numerical approximation of solidification process is resolved on the basis of Equation (2).
(2)$$V_{F_{i,j,k}}\cdot \frac{{\left(\rho I\right)}_{ijk}^{n+1}-{\left(\rho I\right)}_{ijk}^{n}}{\sigma t^{n+1}}=X_{ijk}^{n}-\underset{faces}{\sum }\frac{Ak}{\sigma }\left(T_{ijk}^{n}-T_{adjacent}^{n}\right)-h_{ijk}WA\left(T_{ijk}^{n}-TW_{ijk}^{n}\right)$$
*T*—fluid temperature;*TW*—die temperature;*h*—heat transfer coefficient;*WA*—an interfacial area;*A*—a cell-face area;*K*—an averaged heat conduction coefficient;*∂*—an appropriate spatial increment.

A series of virtual experiments was prepared based on the 3D CAD models. First, a simulation was run for the initial design of the technology. The boundary conditions were adapted to the conditions of the actual high-pressure die casting process. Figure 2 shows the SolidWorks 2011 CAD models of the die and the stirrup casting.

Boundary conditions used for determination of the casting process include individual process steps, their duration, heat transfer coefficients and temperature on the mould surface. A single cycle includes filling the mould cavity, solidification time, opening the die mould and removing the casting, then coating the mould with a release and cooling agent, and finally closing the die mould. Table 1 summarizes the working cycle of the pressure machine used in the simulation model. Initial condition for the die mould is *T_f_* = 180 °C; for the alloy EN AC-47000—initial temperature *T_in_* = 690 °C; and the parameters for velocity in the first phase *V*_1_ = 0.35 m/s, and in the second *V*_2_ = 3.5 m/s. The pressure conditions were set to the atmospheric pressure of 1 bar. 

The simulation of the working cycle of the pressure machine, and more specifically the thermal load of the mould, is used as one of the boundary conditions. The data obtained from the simulation of the cyclic operation of the pressure mould—TDC (thermal die cycling), i.e., the mould temperature profile, allow for a more accurate representation of the actual conditions of high-pressure die casting. The 40 cycles were simulated to determine the stabilization of the mould temperature profile. The visualization of the operation of the pressure mould is shown in Figure 3, which shows the temperature profile in the cross-section of the mould during operation. The chart shows the maximum temperature in the volume of the die mould, which allows determination of the moment of thermal stability.

The assumed boundary conditions showed that the thermal stability of the mould was achieved after 950 s (Figure 4). In the performed experiment, the assumed one work cycle is 26 s, which indicates that the stability was achieved in the 36th cycle.

In the analyzed example, the pressure mould is not complicated. HPDC die moulds can have complex cooling systems, moving cores, ejectors, etc. An additional advantage of TDC analysis is the ability to capture areas that are excessively heat loaded. In the case of a stirrup cast, the characteristic point is the cross-section in the area of the base. The portion of metal that reproduces this part of the casting causes heat accumulation and overheating of the pressure mould in this area (Figure 5).

In the case of very complex mould structures, the previously described method of analysis allows identification of areas in which temperature gradients may occur, causing conditions of thermal stress. This allows for the introduction of additional cooling channels at the mould design stage. The next step includes the process of filling the mould cavity with liquid metal on which force is exerted by the moving piston. The thermal profile determined in the TDC simulation was used for the simulation. The flow of liquid metal through the feeding system is very dynamic. Very high speed in the gating system can reach up to 70 m/s. It can lead to intensive mixing of the liquid melt with air that is in the mould cavity [13]. The filling process is presented in the Figure 6.

Analysis of the flow of the liquid metal shows the flow in a counterclockwise direction. Figure 7 shows the temperature difference (red color—higher temperature, orange color—lower liquid metal temperature) on the fronts of the liquid metal streams, in the area of the feed gap.

Computer simulation allows for a simplified visualization of air evacuation from the die mould. The simplification is related to a computational algorithm that only analyzes what happens to the air in direct contact with the free surface of the liquid metal. More precisely, the calculations require the analysis of two media, liquid metal and gas, which requires a very large calculation effort. Practice shows that simplification allows the capture of errors that occur in the casting technology. Figure 8 shows the visualization of the evacuation of the air volume. There is a visible movement of the air volume caused by the movement of the liquid metal, which is heading towards the overflows where there is a boundary condition defining the area of the vents.

Filling simulation analysis shows (Figure 8) where the air volume is trapped during the flow of liquid metal. That result can be used to predict the defect area where a discontinuity in the structure can occur. Next is the simulation of the solidification of liquid metal. Figure 9 shows the solidification path of the liquid metal in the mould. It shows the successive stages and direction of solidification. It allows for the determination of the last solidifying areas, thus showing the hot spots in the casting.

The next stage of the optimization process to improve the quality of castings with the use of computer simulation consists in determination of the boundary conditions reflecting the actual process parameters. Figure 10 presents the casting with defects occurring in the original configuration.

The presented analysis and selected process parameters reflect the occurrence of predicted defects in the simulation. Figure 11 shows the visualization of solidification areas and defects occurring in these areas.

The defects presented in Figure 10 and Figure 11 are a result of solidification and entrapment of air bubbles. These defects can be distinguished from each other by their size and surface. In the case of porosity, the void walls are covered with formed dendrites, creating a rough surface, while the gas porosity is characterized by a spherical shape with smoothed void walls [14]. Figure 10a shows the filling defect of cold shot. It was formed as a result of the merging of two streams with different front temperature (Figure 7). Figure 12 presents the concept of changing the method of filling the mould cavity. The changes concern the position and size of the gating to achieve optimized filling pattern. 

One of the goals of the optimization is to minimize the disturbances of the free surface of the liquid metal during filling the cavity of the pressure die mould. The visualizations presented in Figure 13 show how changes in the location and size of the gate affect this parameter. In versions 3 and 4 there is a significant reduction in the surface defect concentration value (less red and yellow areas), in comparison with versions 1 and 2 (more red and yellow areas). The configurations and the size of the gate primarily direct the liquid metal stream optimally and limit the formation of the temperature difference on the front. Another analysis involves the direction of the solidification path. The rate of heat dissipation for metal moulds is from 15 to 100 °C/s; for example, cooling in sand moulds is on the order of 1.5–5 °C/s [15]. Premature solidification may also be a casting defect when streaks or flow lines are visible on the casting surface. In the case of the analyzed casting, the solidification did not occur prematurely. Figure 14 shows the solidification path for version 4, which was designated as the final version. This structure is characterized by a directional solidification character, ranging from smaller to larger cross-sections. The last area is the base of the stirrup.

As mentioned above, the solidification in the pressure die is rapid. In the analyzed model, the casting solidifies at t_c_ = 3.5 s, while the biscuit solidifies after t_b_ = 12 s. It should be noted that the casting has very large cross-sections of the base. Due to the geometric shape, this is the area of hot spot. Figure 13 shows the solidification steps. The marked areas solidify last. In these areas, defects of shrinkage can be expected due to the shortage of liquid metal during the change from liquid to solid. Figure 15 shows the porosity prediction in the stirrup casting.

Porosities are located in the areas of hot spot, identified in the solidification analysis. This is in line with the solidification theory that discontinuities occur in the structure in an area where there is a lack of liquid metal. Comparison of the initial casting design and after optimization shows that there is a change in the formation of shrinkage defects. There is no effect of merging the fronts of the liquid metal, causing the formation of weld seams. Additionally, the area where the stirrup is connected to the stirrup leather is free from defects.

### 2.2. Simulation of Exploitation Conditions

The results of the predicted defect were used in the ANSYS 2020R2 software to determine their influence on the casting strength. The location and size of the defects were directly exported as an *.stl file from the Flow3D and prepared as a stirrup model with defects for the ANSYS 2020R2 program. The CAD solid consists of the output model and defects which subtract the void from the solid body by a boolean operation. Overflows, gating and biscuit are removed from the CAD model for calculations (Figure 16).

Computer simulations of exploitation conditions are currently an integral part of the design process. Being properly included in the logical sequence, from the concept through the assessment of production and working conditions, allows for a significant shortening of the designing process and for avoiding defects that may occur in the production process. The ANSYS program uses the finite element method—FEM. To put it very simply, this method consists in solving differential equations in the area of discretization by mathematical approximation [16].

The analysis of forces and reactions acting on the rider’s lower limb allows determination of the maximum values of the stress on the musculoskeletal system of the leg during the assumed riding conditions. The exact characteristics of the acting loads are determined by the biomechanical model of the knee joint described by the Maquet model. The model describes the external and muscular forces and moments in the human lower limb for different ways of mounting a horse, and the reactions transferred from the stirrup to the rider’s foot. The standard rider’s foot inclination angle in the stirrup should be about 12°. According to the assumptions adopted in the Maquet model, the maximum value of the reduced reaction force acting on the rider’s foot should not exceed 100 N, assuming the rider’s seated position. 

In the analyzed casting model, for the stirrup loads, the rider’s seat system in the horse’s jump was adopted as a system of external loads with the maximum values of the forces acting on the stirrup. In the conducted numerical tests, the moment of landing a horse with a rider after a jump over an obstacle from a height of 1.5 m was assumed. The delay time was set at 0.5 s. The rider’s weight should be as low as possible due to the load on the horse. As a boundary condition for the loads acting on the stirrup, it was assumed that the rider’s weight would be 100 kg. The assumed rider’s weight was evenly distributed over each of the stirrups. The force of 500 N was set as the stirrup load. Standard gravity acting on the analyzed element was taken into account, but the direction of the loading force was deviated in the local coordinate system by 12° from the vertical axis, in accordance with the assumed model of riding a horse by a rider. The scheme of the assumed loads and the restraint model are shown in the Figure 17.

The loading force was applied in the central part of the footing (base), while the fastening was assumed in the upper part of the stirrup as a permanent restraint (B). The preliminary analysis of static loads indicated the stress areas of the stirrup structure; however, the maximum stress values do not exceed the value of 45 MPa. The distribution of von Mises stress fields is shown in Figure 18, while the reduced displacements are shown in Figure 19.

The critical values of reduced stresses are located in the upper part of the stirrup, in the area of the stirrup pad attachment. Another dangerous area of the casting is the narrowing of the stirrup structure in the area of mounting the shock absorbers. Therefore, attention should be paid to the quality of the casting in these areas in order to guarantee no defects in casting. The maximum deformation of the stirrup structure under static loads is about 0.2 mm. For the assumed aluminum alloy, it is a safe value and much lower than the minimum elongation. The areas of predicted defects are located in the lower part of the casting—one defect in the central part of the footing and two small defects symmetrically arranged in the side arms of the lower part of the stirrup, i.e., in places of non-constricted structures. Therefore, it should be assumed that the porosity in these areas should not affect the strength of the stirrup structure. The modeled construction of the stirrup casting with defects was also subjected to a load analysis in order to determine the impact of these defects on the accumulation of critical stresses. The obtained results confirmed the assumed conclusion. Both the maximum values of stresses and displacements in the stirrup casting are at a similar level as in the casting without defects. Also, the analysis of normal stresses did not show any accumulation of material stress in the area of defects.

In order to determine the influence of the dynamic load on the stirrup on the development of destructive stresses, analyses were carried out according to the damping scheme of the acting forces in 0.5 s. The scheme of the stirrup structure’s construction effort, taking into account the damping of the structure material, is shown in Figure 20, while the analysis time was limited to 0.3 s.

The dynamic analysis showed a very similar character to the occurrence of the maximum stress values. The maximum values of the stresses slightly exceed the value of 115 MPa, with the global deformation of the lower part of the stirrup being approximately 0.35 mm. The maximum load values were determined for the time *t* = 0.05 s.

On the basis of the obtained values of the stress of the structure, which are presented in Figure 21, it can be concluded that the critical area of the casting is located in the upper, narrowed part of the casting. There is no shrinkage porosity in this area; however, due to the design of the casting, it is the critical area most exposed to loading. The analysis of the stress distribution of the stirrup structure allowed an estimation of the safety coefficient of the stirrup structure material which is presented in the Figure 22.

The critical value of the safety factor (determined for the time *t* = 0.05 s) in this area is approximately 1.3 (determined for the time *t* = 0.05 s) for the dynamic nature of the stirrup loading. This value, with a stabilized load pattern, reaches the value of about 3.47. The analyzed area, in the case of a variable nature of the acting loads, may be the initiator of a casting fracture process. However, it should be stated that the intended use of the stirrup casting, due to its dressage character, should guarantee safe working conditions for the casting.

### 2.3. Material Analysis

Additionally, in order to verify the technological changes, a visual analysis of the microstructure was carried out in selected parts of the casting made on the basis of the initial and modified technology. Figure 23 shows the sampling areas ABC for chemical composition analysis, which were the same for both technologies. The samples themselves were cut by cutting with water.

The process of filling and solidification with almost every casting method is associated with the formation of defects. In the case of high-pressure casting, the direction of the solidification is very difficult. The most common defects are: shrinkage porosity, gas porosity and leakage. The porosity is formed in the area of hot spots and is characterized by jagged edges that constitute dendrites. In the case of gas porosity, the surface of the discontinuity is smooth. This is directly due to the air bubble which is closed by the flowing liquid metal. The last leakage defect may result from a combination of previous defects that create a discontinuity in the structure passing through the entire section of the casting wall. Besides the geometry, very important is the chemical composition of the alloy. A chemical composition test was carried out on a stationary GNR mL300 spectrometer. The chemical composition of the alloys is presented in Table 2. The initial casting (OMI) and the new casting (FRI) were analyzed.

In addition to changing the shape of the gating system, a different casting alloy was used. The former eutectic alloy was converted to an around-eutectic alloy containing the addition of Fe and Mn. Such composition improves the casting properties, e.g., castability, thanks to which the complete filling of the mould cavity and the reduction of weld defects are ensured. For cognitive purposes, EDS (energy dispersive spectroscopy) research was conducted. Figure 24 shows the areas of analysis for individual alloys that are in the as-cast condition.

This study shows that the alloy composition for both tested materials can be classified as alloys of the EN AC-43400 and EN AC-4700. The mechanical properties for these alloys, according to the EN 1706: 2010 standard, are presented in Table 3. Mechanical properties were not tested in the course of the work.

The microstructure components in the samples analyzed with the EDS method show the existence of basic components for AlSi alloys. Here there is α-Al (Figure 24 FRI-Spot 7, OMI-Spot 2), Si eutectic (Figure 24 FRI-Spot 3 and 8, OMI-Spot 4) and Mg2Si (Figure 24 FRI-Spot 2, OMI-Spot 5). Differences in the microstructure resulting from the content of Mn and Cu. In the FRI alloy, the Cu content is 0.04%, and Mn—0.25%, while in the OMI alloy, the Cu content is 0.21%, and Mn—0.06%. In the case of the FRI alloy with the addition of Mn, there is a phase α-Al_15_ (Fe, Mn)_3_Si_2_ (EDS, Figure 24 FRI Spot-5 and 6). The addition of manganese facilitates the transformation of the iron-rich Al_5_FeSi phase into Al_15_(MnFe)_3_Si_2_, which was observed in the EDS studies. Due to the addition of Mn, the alloy increases the strength properties. However, in the alloy with the addition of Cu, there is the so-called Chinese script, α-AlFeSiCuMg, which is undesirable in a casting due to its deterioration in mechanical properties.

## 3. Results

Improving the quality of products allows the reduction of production costs by eliminating defects through analyzing the current production conditions. The principle of optimization is not limited to the one chosen, but is general for all branches of manufacturing single parts and entire complex mechanisms and devices. The presented case of the equestrian stirrup shows that the simulations can help investigate and describe occurring defects and limit their occurrence by proposing various technological variants without the real manufacturing process. A very important issue in the designing of casting technology is the assessment of the nature of the flow of liquid metal and the temperature in the volume of the liquid alloy. In the case of the initial technology, the excessive temperature drop caused defects in the form of cold shots in the area around which the stirrup is attached to the saddle through the stirrup leather. Changing the direction of the liquid metal flow ensures more closeup filling of the liquid alloy and solidification. The FEM analysis carried out for the stirrup with a defect in the form of porosity showed that the location does not affect the safety of the casting. Only in the area of the transition from the column to the curved part is it exposed to higher stress values. The values it achieves are not in the plastic range of the material characteristics for static (max. stress 45 MPa) and dynamic (max. stress 115 MPa) conditions. The danger of element destruction in the form of weakening the structure in this area has been eliminated by changing the gating technology.

## 4. Conclusions

The use of computer simulations helped in designing an optimal gating system through the analysis of the liquid metal flow pattern and solidification path in the initial design. Changing the shape of the gating system allowed elimination of the danger of the formation of cold shots. Introducing the defects in the CAD model for exploitation analysis allowed assessment of its influence on the strength of the casting. Additionally, the application of a different alloy with higher mechanical properties than the original one ensured high quality of the part.

## Figures and Tables

**Figure 1 materials-15-06781-f001:**
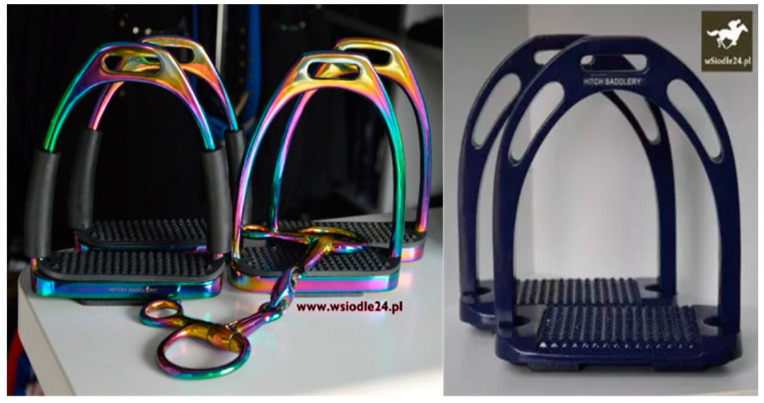
Commercially available stirrups [10]. Source: www.wsiodle24.pl. Reprinted/adapted with permission from Ref. [10]. Copyright www.wsiodle24.pl.

**Figure 2 materials-15-06781-f002:**
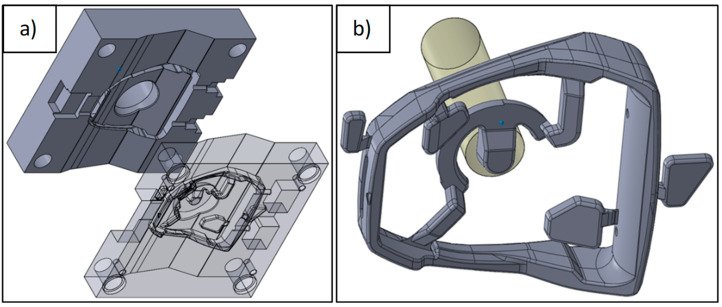
CAD model (**a**) die mould and (**b**) stirrup casting with overflows and gating system.

**Figure 3 materials-15-06781-f003:**
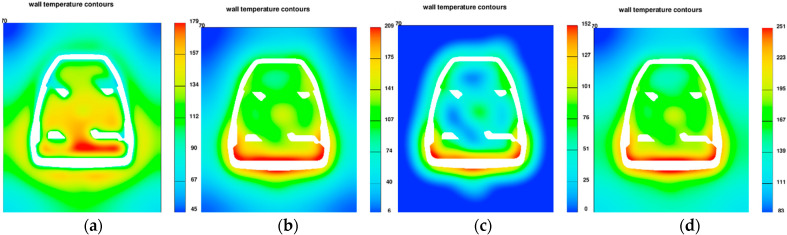
Visualization of the cyclic heating and cooling of the mould during steps presented in Table 1, [°C]: (**a**)—filling, (**b**)—solidification, (**c**)—ejection and spraying, (**d**)—closing.

**Figure 4 materials-15-06781-f004:**
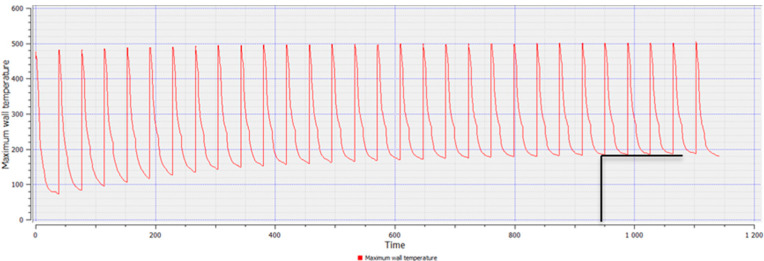
Temperature curves during the cyclic heating and cooling of the mould during casting process; the temperature stabilization point is marked by black line.

**Figure 5 materials-15-06781-f005:**
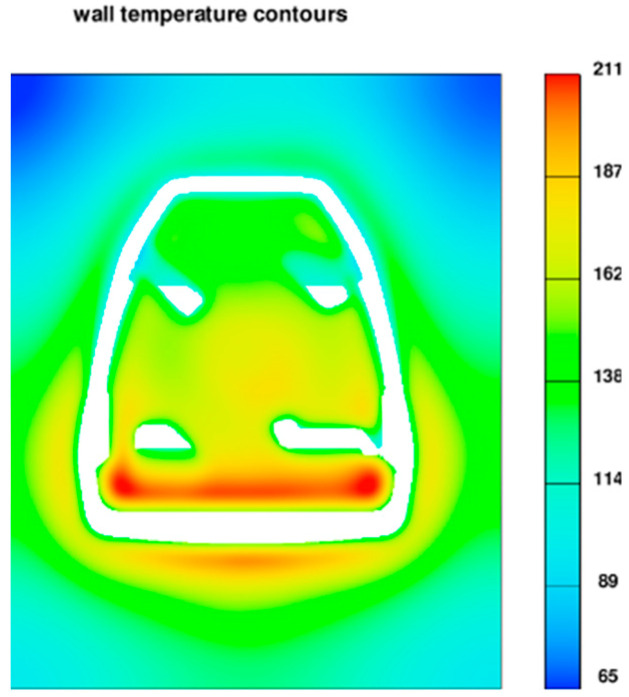
The area of intense heat accumulation (211 °C) in the die mould, cross-section view.

**Figure 6 materials-15-06781-f006:**
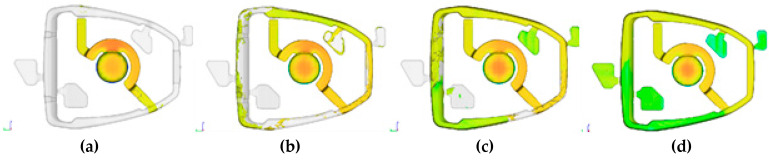
Visualization of the temperature range (red color higher, green lower value) of the liquid alloy during filling process time—(**a**) 0.009 s., (**b**) 0.0014 s., (**c**) 0.025 s., (**d**) 0.052 s.

**Figure 7 materials-15-06781-f007:**
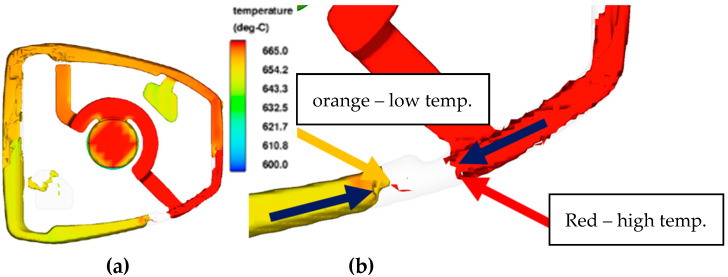
Visualization of the temperature (range from 680 °C to 640 °C) of the liquid metal, (**a**) whole casting, (**b**) enlarged view.

**Figure 8 materials-15-06781-f008:**
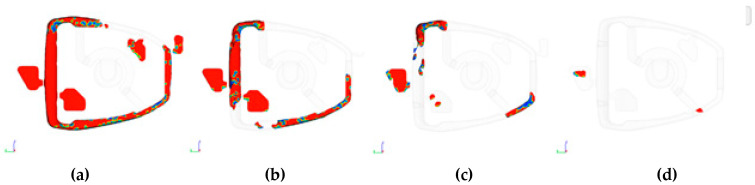
Visualization of the evacuation of the air volume (air bubble is visible) from the mould cavity, (**a**) 0.009 s., (**b**) 0.0014 s., (**c**) 0.025 s., (**d**) 0.052 s.

**Figure 9 materials-15-06781-f009:**
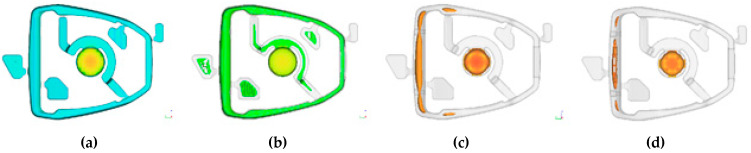
Visualization of the solidification path (visible from liquid to solid) of the liquid metal—time frame (**a**) 0.2 s., (**b**) 0.5 s., (**c**) 1.7 s., (**d**) 2.7 s.

**Figure 10 materials-15-06781-f010:**
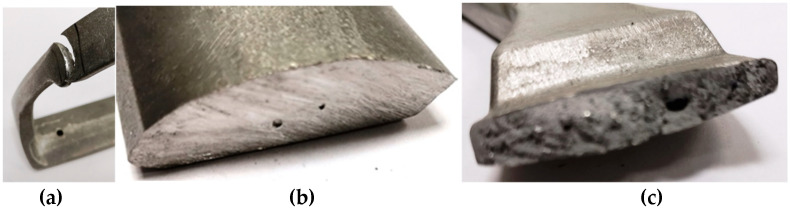
Defects of (**a**) cold shut, (**b**) air bubble, (**c**) shrinkage porosity in the stirrup casting, initial configuration.

**Figure 11 materials-15-06781-f011:**
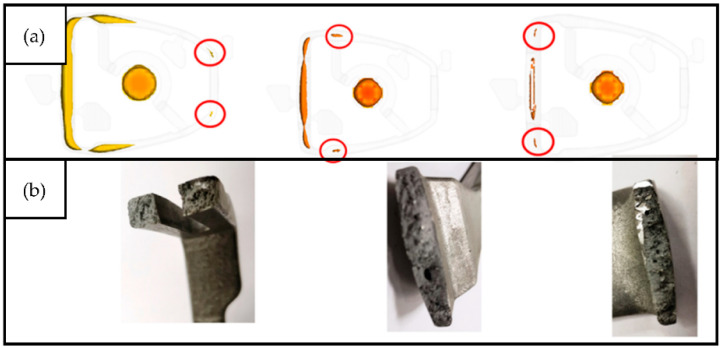
Visualization of the solidification path and defects in the casting (**a**) solidification process, (**b**) casting defects.

**Figure 12 materials-15-06781-f012:**
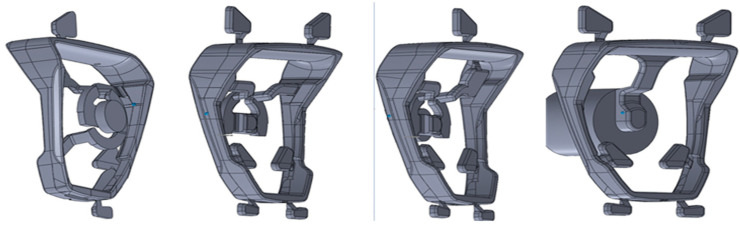
Examples of variants of the casting technology of the stirrup casting.

**Figure 13 materials-15-06781-f013:**
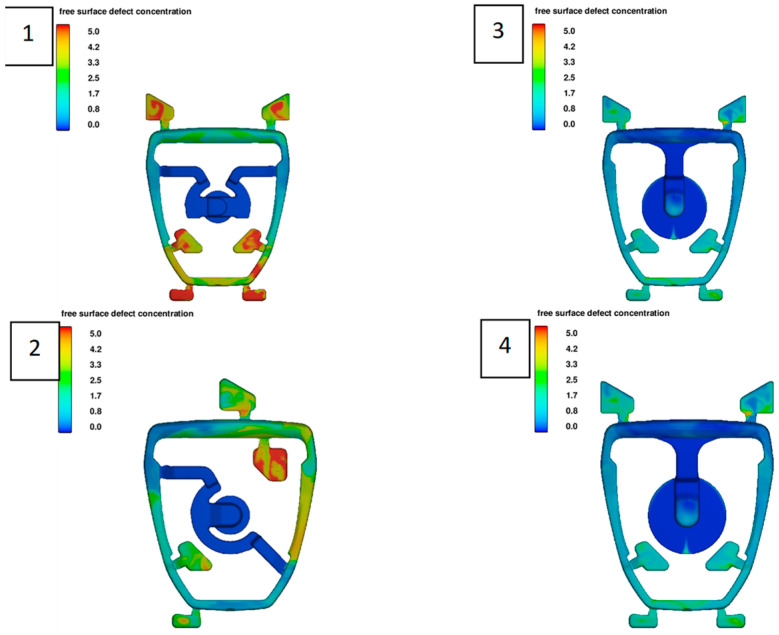
Comparison of the surface defect concentration in chosen casting technologies (**1**–**4**).

**Figure 14 materials-15-06781-f014:**
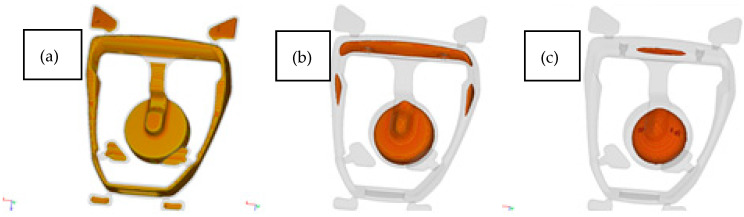
Solidification path of the liquid metal in the cavity (visible liquid fraction) time frame—(**a**) 0.3 s., (**b**) 1.6 s., (**c**) 3.7 s.

**Figure 15 materials-15-06781-f015:**
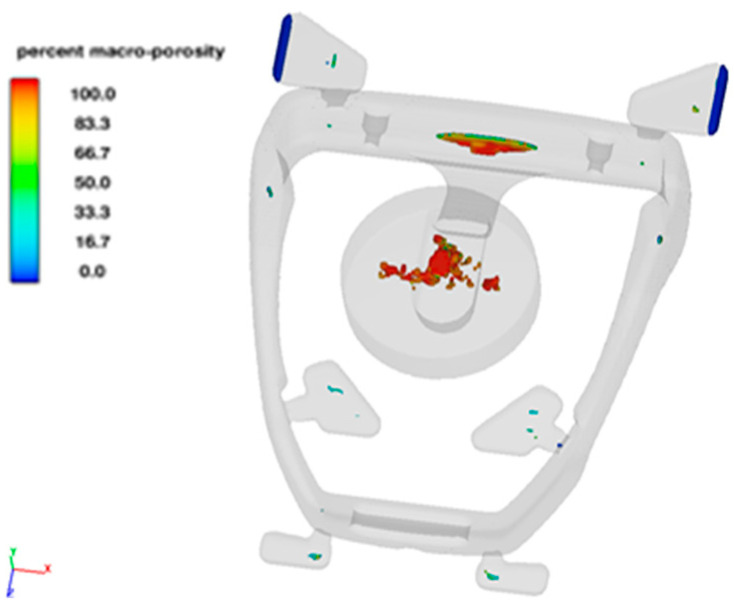
Prediction of porosity in the stirrups casting—technology version 4.

**Figure 16 materials-15-06781-f016:**
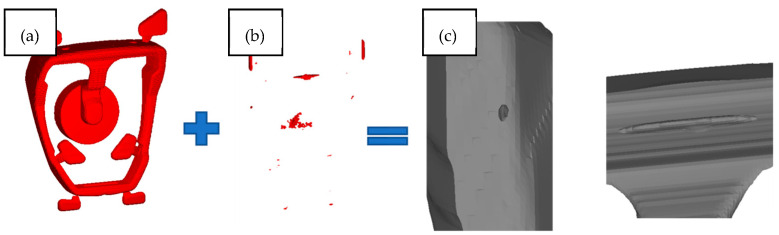
Preparation of the CAD model for ANSYS simulations taking into account (**a**) the shape of the stirrup and (**b**) the location of defects obtained in the simulation of the solidification process; (**c**) final geometry with defects.

**Figure 17 materials-15-06781-f017:**
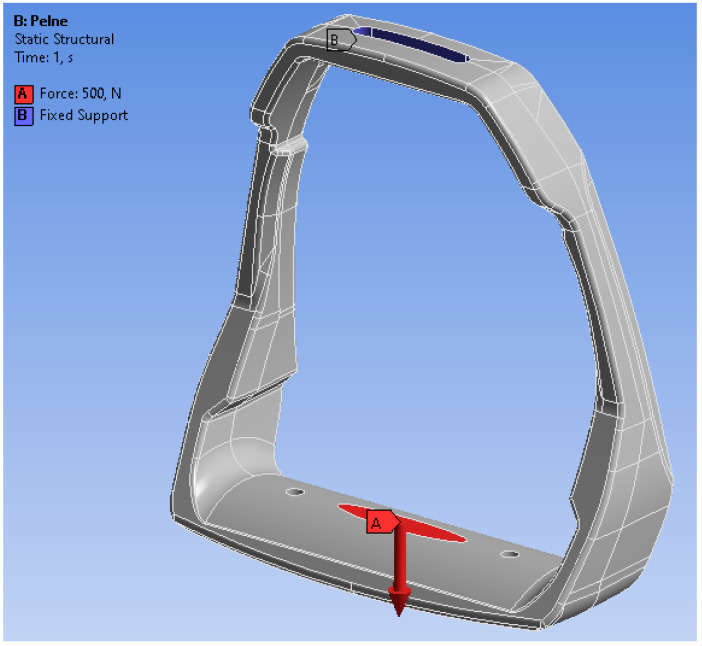
The model of boundary conditions adopted in the analysis of the loads of the stirrup casting, A—means the direction of the force.

**Figure 18 materials-15-06781-f018:**
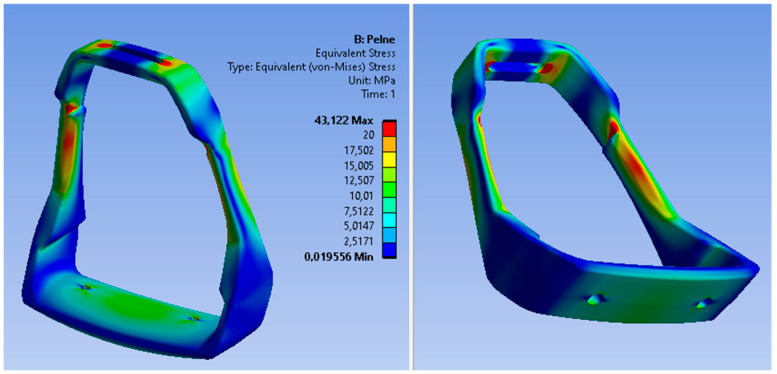
Stress of the stirrup structure under static loads.

**Figure 19 materials-15-06781-f019:**
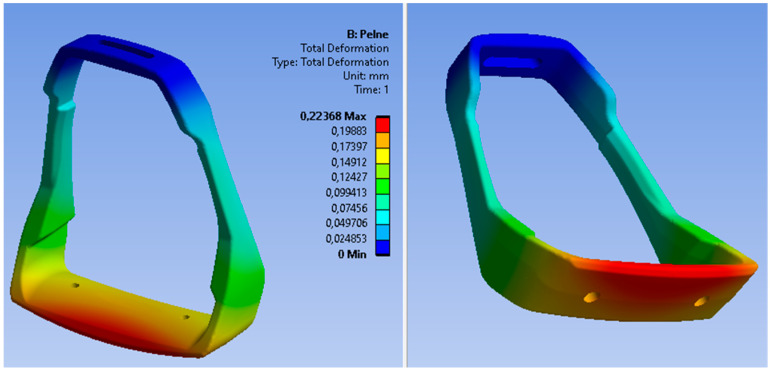
Deformation of the stirrup structure under static loads.

**Figure 20 materials-15-06781-f020:**
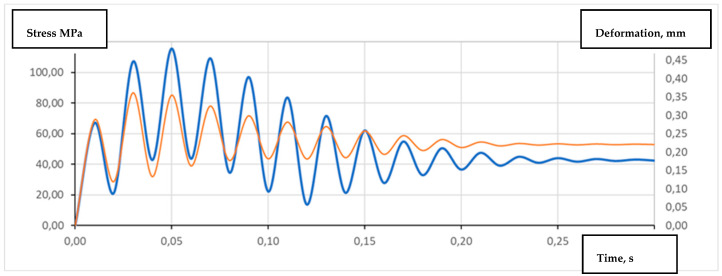
Characteristics of stresses and deformations in the dynamic loading of stirrups.

**Figure 21 materials-15-06781-f021:**
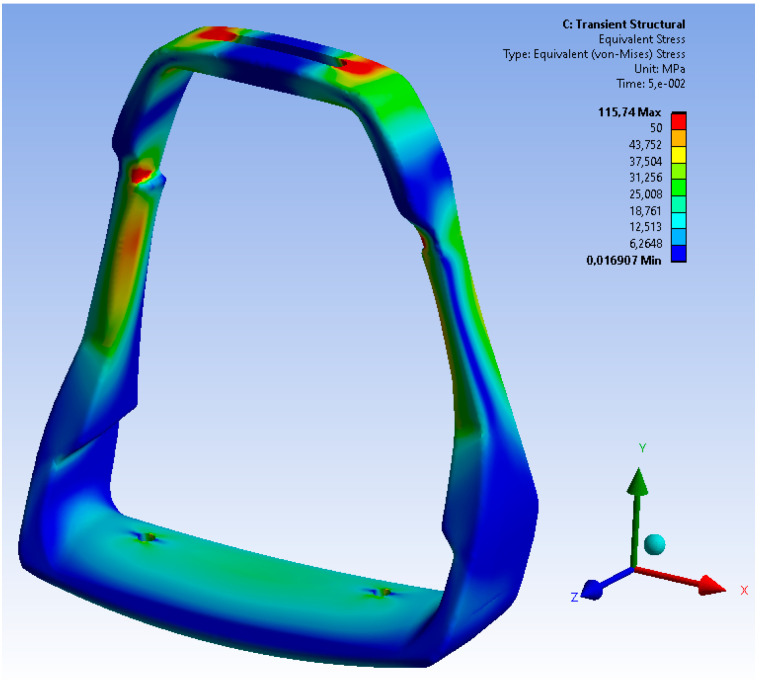
Stress of the stirrup structure under dynamic loads (for time *t* = 0.05 s).

**Figure 22 materials-15-06781-f022:**
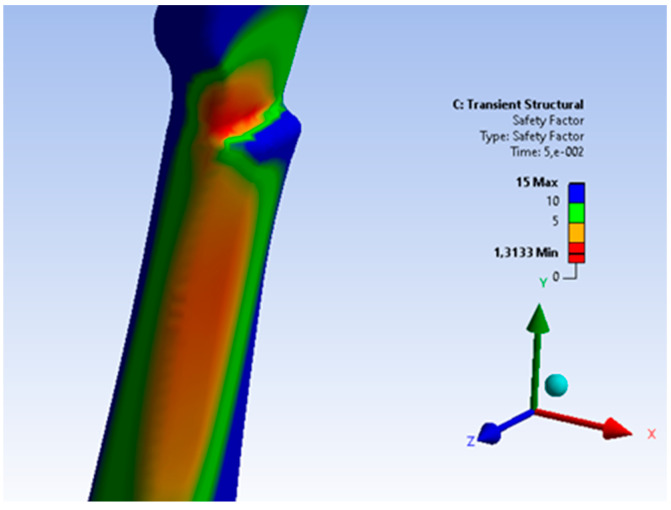
Estimated distribution of the safety factor for the dynamic nature of the acting loads (time *t* = 0.05 s).

**Figure 23 materials-15-06781-f023:**
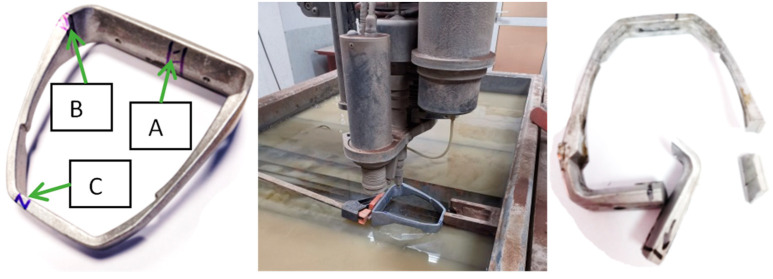
Area of sampling from castings.

**Figure 24 materials-15-06781-f024:**
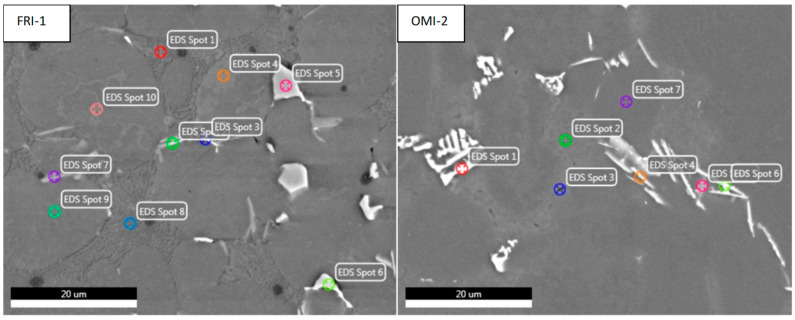
Map of the analysis of elements occurring in the phases for FRI and OMI alloys.

**Table 1 materials-15-06781-t001:** The working cycle of the pressure mould.

No.	Step	Duration [s]
1.	Solidification	14
2.	Opening the mould and removing the casting	13.5
3.	Cleaning and spraying	26
4.	Closing the mould	5

**Table 2 materials-15-06781-t002:** Chemical composition of castings.

Name	Si	Fe	Mg	Cu	Mn	Ti	Cr	Al.
FRI–1	8.92	0.57	0.26	0.04	0.25	0.08	0.01	Bal.
OMI–2	11.35	0.48	0.18	0.21	0.06	0.07	0.01	Bal.

**Table 3 materials-15-06781-t003:** Properties of the alloys.

Alloy	UTS Rm, MPa	Rp02, MPa	Elongation A, %	Hardenss HB
EN AC-43400 (Fri)	240	140	1	70
EN AC-47000 (OMI)	180	97	1.7	60

## Data Availability

Data are contained within the article.

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
