# Peer review of "Technological Optimization of the Stirrup Casting Process with the Use of Computer Simulations"

_materials, 2022, doi:10.3390/ma15196781_

Round 1

Reviewer 1 Report

This paper simulated flow and solidification behavior of equestrian stirrups during high-pressure die casting processing using FEM. It is an interesting work. But the paper is not readable and the logicality is not good, and some typos and inappropriate sentences are found. Overall, in the present form it cannot be accepted for publication.

--Section 1 introduction: please add some references about casting process simu;ation;

--Line 93,96,105 and 106: please replace eq.1 and eq.2 for (1) and (2), also give the explanation of the subscripts in eq.2

--Line 120-125: Please give the values of simulation parameters, such as material, temperature, pressure and so on;

--Figs is not clear, please revise them and give a caption for every figure and explain them. There is the same problem of Fig.4-9,Fig11,and son on, please revise them.

--What defect occur in original configuration and indicate the defect in Fig.10;

--Line 227 and 223: 。c/s should be ºC/s;

--Line 221-223, what is version 3 and 4, version 1 and 2?

-- how to understand Fig 16, please present it in detail.

--please add the other section title for “ section 2. Materials and Methods” in order to have better readability;

--please analyze the microstructure distribution characteristic in Fig.24 and discuss them.

Author Response

--Section 1 introduction: please add some references about casting process simuation;

Added literature regarding high pressure die casting.

--Line 93,96,105 and 106: please replace eq.1 and eq.2 for (1) and (2), also give the explanation of the subscripts in eq.2

All given notations in Flow3D guide are included in the manuscript.

--Line 120-125: Please give the values of simulation parameters, such as material, temperature, pressure and so on;

Initial conditions for die mould is Tf = 180 oC, the alloy EN AC-47000 – initial temperature Tin=690 oC, and parameters for velocity in 1st phase V1 = 0,35 m/s and 2nd V2 = 3,5 m/s. The pressure conditions were set to the atmospheric pressure of 1 bar.     

--Figs is not clear, please revise them and give a caption for every figure and explain them. There is the same problem of Fig.4-9,Fig11,and son on, please revise them.

Figures was revised,

--What defect occur in original configuration and indicate the defect in Fig.10;

--Line 227 and 223: ã€‚c/s should be ºC/s;

--Line 221-223, what is version 3 and 4, version 1 and 2?

-- how to understand Fig 16, please present it in detail.

Description of the fig is in text above “The CAD solid consists of the output model and defects which subtract the void from the solid body by a boolean operation. Overflows, gating and biscuit are removed from the CAD model for calculations. A pictorial process of assembling geometry is presented in Fig. 16."

Additionally figure caption was changed

--please add the other section title for “ section 2. Materials and Methods” in order to have better readability;

Numbering and section signatures have been added

--please analyze the microstructure distribution characteristic in Fig.24 and discuss them.

Since the EDS was done and described do optical microscope microstructure was removed.

Reviewer 2 Report

After correcting the mentioned objections, the manuscript is acceptable for this journal.

Author Response

All sections of the articles have been changed in terms of language, style and content.

Round 2

Reviewer 1 Report

Authors corrected some mistakes and added some explanations, the manuscript was much improved. However,some inappropriate sentences are still found. In my opinion, it cannot be accepted for publication in the present form.

Line 57-63: These sentences need to be corrected in order to be better readability;

Fig.3, Fig.5, Figs.6-9, Fig.11-14 and the other similar figures are consist of some figures, but are short of captions like Fig.10, please give a caption for every figure and explain them in the text.

Author Response

Authors corrected some mistakes and added some explanations, the manuscript was much improved. However,some inappropriate sentences are still found. In my opinion, it cannot be accepted for publication in the present form.

Language mistakes was improved.

Line 57-63: These sentences need to be corrected in order to be better readability;

Computer simulations of high pressure die casting process includes process parameters i.e. velocity profile of the plunger, thermal profile of the die mould, ventilation and cooling. Research work describing mentioned conditions [16-18] shows how computer simulations can help solving process issues. For example, very important parameters describing the liquid metal velocities can be optimized by pQ2 chart. Additionally the influence of air evacuation is presented as a very important parameter of whole casting method. Optimization of velocity factor additionally allows for lowering the costs. Better knowledge of process parameters helps to produce high quality castings.

Fig.3, Fig.5, Figs.6-9, Fig.11-14 and the other similar figures are consist of some figures, but are short of captions like Fig.10, please give a caption for every figure and explain them in the text.

Fig. 3 Visualization of the cyclic heating and cooling of the mould during steps presented in the table 1, [oC] a – filling, b – solidification, c – ejection and spraying, d – closing

Fig. 5 – one figure was deleted since the area of accumulation of heat can be presented on one figure

Fig. 6 Description of the time frames was added

Fig. 7. Actual description “Visualization of the temperature (range from 680 oC to 640oC) of the liquid metal” – tells that the fronts can cause cold shot defect, and later is in the text “Fig. 10a shows the filling defect of cold shot. It was formed as a result of merging of two streams with different front temperature (Fig. 7).”

Fig. 8 The time frame of each step is added in the description – and description in the text “Filling simulation analysis shows (Fig. 8) where the air volume is trapped during the flow of liquid metal. That results can be used to predict the defected area where a discontinuity in the structure can occur.”

Fig. 9 Actual description “Visualization of the solidification path (visible from liquid to solid) of the liquid metal (after 0,2 s.; 0,5 s.; 1,7 s., 2,7 s.)” in my opinion is sufficient – description in the text “The defects presented in Figs. 10 and 11 are a result of solidification and entrapment of air bubbles. These defects can be distinguished from each other by their size and surface.”

Fig. 12 Actual description “Examples of variants of the casting technology of the stirrup casting” – and the description in text Fig. 12 presents the concepts of changing the method of filling the mould cavity. The changes concern the position and size of the gating to achieve optimized filling pattern.

Fig. 13 Actual description “Comparison of the surface defect concentration in chosen casting technologies (1-4)” and the description in the text The visualizations presented in Fig. 13 show how changes in the location and size of the gate affect this parameter. In versions 3 and 4 there is a significant reduction in the surface defect concentration value (less red and yellow areas), in compare to versions 1 and 2 (more red and yellow ares). The configurations and the size of the gate primarily direct the liquid metal stream optimally and limit the formation of the temperature difference on the front.

Fig 14. Time frames of each step is added